# Influence of Loads and Loading Position on the Muscle Activity of the Trunk and Lower Extremity during Squat Exercise

**DOI:** 10.3390/ijerph192013480

**Published:** 2022-10-18

**Authors:** Ju-Hyung Park, Seung-Jea Lee, Ho-Jin Shin, Hwi-Young Cho

**Affiliations:** 1Department of Health Science, Gachon University Graduate School, Incheon 21936, Korea; 2Department of Medical Sciences, Soonchunhyang University, Asan-si 31538, Korea; 3Department of Physical Therapy, Gachon University, Incheon 21936, Korea

**Keywords:** high bar back squat, low bar back squat, lower extremity, trunk muscle

## Abstract

This study aimed to investigate the effect of the load and bar position on trunk and lower extremity muscle activity during squat exercise. High bar back squats (HBBS) and low bar back squats (LBBS) were performed in random order at 50%, 60%, and 70% loads of one repetition maximum by 28 experienced healthy adult men who had been performing squats for at least one year. Before the experiment, the maximal voluntary contraction of the vastus medialis, vastus lateralis, rectus femoris, biceps femoris, rectus abdominis, transverse abdominis, external oblique, and erector spinae muscles was measured by means of surface electromyography. In addition, eccentric and concentric exercises were performed for 3 s each to measure the muscle activity. There was a significant difference in muscle activity according to the load for all muscles in the eccentric and concentric phases (*p* < 0.05), indicating that muscle activity increased as the load increased. In addition, in the comparison between HBBS and LBBS, significant differences were shown in all lower extremity muscles and all trunk muscles except for the external oblique in the concentric phase according to the bar position (*p* < 0.05). HBBS showed a higher muscle activity of the lower extremity in the eccentric and concentric phases than in LBBS, while LBBS showed a higher muscle activity of the trunk muscle in the eccentric and concentric phases than in HBBS (*p* < 0.05). HBBS requires more force in the lower extremity than LBBS and is particularly advantageous in strengthening the muscular strength of the quadriceps. In contrast, LBBS requires more muscle activity in the trunk than HBBS and is more effective in carrying heavier loads because of the advantage of body stability. This study suggests that rehabilitation experts apply the bar position and load as important variables affecting the intensity and method of training for target muscle strengthening of the lower extremities and trunk.

## 1. Introduction

In modern society, people take more interest in health due to economic leisure and improvements in living standards; therefore, the participation rate in leisure and sports activities is increasing. According to the Korean National Life Sports Survey, the participation rate of adults in sports activities increased from 45.5% in 2013 to 60.1% in 2020. Among various sports activities, swimming (32.6%), weight training (22.7%), and yoga–pilates–taebo (19.9%) showed the highest participation rate [1].

Weight training consumes more energy than aerobic exercise and by using auxiliary tools (such as dumbbells) and various weight-training equipment, strength and muscular endurance can be strengthened, and muscle power can be improved owing to the resistance exercise [2,3]. In addition, appropriate strength training can reduce the feeling of helplessness that can be caused by aging, and preserve the functioning of muscles through skeletal and muscle strengthening [4,5].

Among various weight-training exercises, squat training is the most commonly used exercise to enhance the muscle strength of the lower limb. This exercise can strengthen the quadriceps femoris, hamstrings, gluteus maximus, and core muscles (transverse abdominis, multifidus, abdominal muscles, erector spinae) [6,7]. Closed chain exercises, such as squat exercise, require more joint movement compared to open chain exercise and are more effective in stimulating proprioception along with stability [5]. In addition, squat exercises not only strengthen muscles but also treat and prevent injuries in patients with musculoskeletal disorders of the lower extremities, such as hip joint pain syndrome and ankle instability [8,9].

While performing squat exercises using a bar, the position of the bar can be changed depending on the purpose of the exercise or the target muscle [10,11]. These squat exercise methods include back squats, overhead squats, and front squats, of which back squats are most commonly performed during weight training in gyms and rehabilitation centers [12,13]. In addition, the back squat method is divided into high bar back squat (HBBS) and low bar back squat (LBBS), according to the difference in bar positions. When performing these two back squat movements, the muscle activity of the trunk and lower extremity is different, and appropriate exercise should be applied to the participant’s functional state accordingly [14,15]. However, most of the studies comparing HBBS and LBBS have reported an effect on the lower extremity muscles, and analysis of trunk muscles according to the change in the inclination of the upper body, is required. Compared with LBBS, HBBS exercises are performed in a more upright posture, and the moment arms of the knee and ankle joints are relatively larger [16]. On the other hand, in LBBS, the trunk is inclined more forward than in HBBS, and the moment arm of the hip joint is larger [17]. This indicates that the degree of trunk bending and the change in the moment arm of the lower extremity joints are related to the muscle activity of the trunk [18].

Until now, many previous studies have reported a difference in the activity of lower extremity muscles depending on the methods of the squat. However, a study that compared trunk muscle activity to the bar position during the squat was limited to the erector spinae [16]. It is required to investigate the effect of the bar position on the activity of various abdominal and core muscles during the squat. In addition, the previous study that investigated the effect of barbell load on the squat was performed under conditions with small differences between loads (60% vs. 65% vs. 70%) [19]. All three load conditions in the study are applied to optimize hypertrophic gains [20]. Training could be used not only for hypertrophic change but also to enhance strength or improve muscular endurance. Therefore, it is necessary to investigate the effect of bar position and load differences on the muscle activation patterns in the trunk and lower extremities during squat performance.

Therefore, the purpose of this study was to investigate the differences in the trunk and lower extremity muscle activity according to the load and bar position during squat exercises.

## 2. Materials and Methods

### 2.1. Participants

This study was conducted over a period of 2 months from October to November 2021, and 28 participants were recruited through a recruitment notice at the G fitness center or gym located in Incheon, Republic of Korea. The participants were healthy men and had been performing squats once a week or more for at least 1 year. In addition, the exclusion criteria were set as follows [13]: (1) participants with pain or fractures in the spine and lower extremities; (2) those with limited range of motion in the spine and lower extremities; (3) those with a history of surgery on the spine and lower extremities; and (4) those with lesions in the central or peripheral nerves. The ethical issues and procedures of the study were approved by the institutional review board of Gachon University (IRB number: 1044396-202 109-HR-202-01).

The sample size was calculated using the G-Power software (G* Power ver. 3.1.9.2; University of Kiel, Aichach, Germany). In this study, the median value of Cohen’s f, 0.25, was used; There was no previous study that had used it as a criterion for the effect size. The sample size was calculated by selecting ANOVA: repeated measures, within-between interaction of F test, effect size f = 0.25, alpha err prob = 0.05, power (1-beta err prob) = 0.8, number of groups = 2, and number of measurements = 3. Therefore, the appropriate sample size was calculated to be 28, and finally, 32 patients were evaluated, considering a dropout rate of 10%.

### 2.2. Procedure

All participants signed a written informed consent prior to the experiment. In this study, information on age, height, and weight, training experience was collected from the participants before the experiment to identify their general characteristics. In addition, the research team fully explained the experimental procedure, purpose, and precautions to all participants. Participants were instructed to restrict lower extremity exercise and get enough sleep and abstinence from alcohol one day before participating in the experiment [21].

The experiment was carried out in a total of two steps with an interval of 3 days [22,23]. In the first step, the one repetition maximum (1 RM) of the participant’s HBBS and LBBS was measured. Before measuring the 1 RM, warm-up was performed for 10 min on a treadmill [24], and the 1 RM value was determined by calculating 10 RM as 75% of 1 RM [2]. Before performing step 2, after measuring maximal voluntary contraction (MVC), six conditions were randomly assigned using “Research Randomizer” software. Eight muscles selected during the squat were measured using an electromyography device. The muscles selected are as follows: vastus medialis (VM), vastus lateralis (VL), rectus femoris (RF), biceps femoris (BF), rectus abdominis (RA), transverse abdominis (TrA), external oblique (EO), and erector spinae (ES) muscles.

### 2.3. Intervention

The participants performed HBBS and LBBS three times each at 50%, 60%, and 70% of 1 RM intensity, and a total of six conditions were performed in a random order. For HBBS, the bar was placed on the upper trapezius, and for LBBS, it was placed between the scapulae on both sides and on the posterior deltoid [16]; the width of the foot was set equal to the width of the pelvis (Figure 1). In addition, when performing squats, the thigh was instructed to be positioned below the horizontal level, and the motion was measured by dividing it into eccentric and concentric phases for 3 s each at a metronome speed of 60 bpm [25]. During squat exercises, the eccentric phase was defined as the highest to the lowest position, and the concentric phase as the lowest to the highest position [19]. The rest period between sets was set to 2 min [13], and a 10-min rest period was provided after the MVC measurement to minimize muscle fatigue. For participants’ safety, a power rack (NEWTECH, ROK, Gimhae-si, Korea) equipped with a safety bar was used.

### 2.4. Measurement

Surface electromyography (sEMG) (BIOPAC MP160; BIOPAC Inc., Goleta, CA, USA) was used to measure the muscle activity of the trunk and lower extremities during HBBS and LBBS. Signals collected through sEMG were processed using the AcqKnowledge 5.0.1 software (BIOPAC systems, Inc., Goleta, CA, USA). A disposable surface electrode was attached after wiping and disinfecting the skin surface with an alcohol swab to obtain accurate data. The electrode was attached to the target muscles as mentioned previously [26,27], and the ground electrode was attached to the patella, fibular head, anterior superior iliac spine, and posterior superior iliac spine.

For raw electromyogram (EMG) signal analysis, the sampling rate was set to 1000 Hz, and band-pass filter was set to 30–500 Hz. The collected EMG signals were subjected to root mean square processing, and the data obtained from each muscle were quantified using MVC. In addition, the MVC of each muscle was measured as per the SENIAM manual and sEMG sensor procedures [28].

To measure the MVC of the TrA, participants were instructed to pull back their belly button while sitting and maintain this posture for as long as possible while exhaling like air released from a balloon. To measure the MVC of the EO muscle, the participant assumed a side-lying position with the lower extremities fixed to a band and subsequently maintained a side-bending position. To measure the MVC of the RA, the participant flexed the knee and fixed the ankle while in the supine position. The participant placed both hands on the chest, lifted and maintained the torso, and the evaluator provided resistance to the trunk. To measure the MVC of the ES, the participant was placed in a prone position with the legs fixed and torso raised. The participant attempted to maintain this posture, and the evaluator provided bending resistance to the trunk. To measure the MVC of the RF, VM, and VL, the participant sat on a table such that the feet did not touch the ground and had the knee flexion between 70° and 90° of the knee bending angle, and the evaluator provided extension resistance to the participant’s shin. To measure the MVC of the BF, the participant was asked to maintain a 90° knee flexion in the prone state. The evaluator applied resistance to the participant’s knee in the direction of the flexion. The measurement was performed three times for 5 s, and a rest time was provided for 1 min after one measurement to minimize the occurrence of muscle fatigue. For the collected EMG signal, the average value for the remaining 3 s, except for 1 s before and after, was used as the data.

### 2.5. Statistics Analysis

SPSS software (version 25.0; SPSS Inc., Chicago, IL, USA) was used for the data analysis, and the mean and standard deviation for each variable were calculated and compared. The Shapiro–Wilk test was used to confirm the normal distribution of the original data. The comparison of muscle activity according to load was conducted using repeated measures analysis of variance and the Friedman test. For post hoc testing, the paired *t*-test or Wilcoxon signed-rank test was used. To compare muscle activity according to bar position, the Wilcoxon signed-rank test was used. All statistical significance levels were set to *p* < 0.05.

## 3. Results

### 3.1. Characteristics of Participants

Among the volunteers who applied for this study, those who met the exclusion criteria were dropped out. As a result, a total of 28 subjects participated in this study. The general characteristics of the participants are shown in Table 1.

### 3.2. Muscle Activity of Lower Extremity

LBBS significantly increased the muscle activity in 60% of all muscles compared to 50% in the eccentric phase; in BF alone, the muscle activity significantly increased in 70% (*p* < 0.05) (Table 2 and Table 3). In the concentric phase, muscle activity significantly increased with increasing load in all muscles except the RF (*p* = 0.430 in 50% vs. 60%) (*p* < 0.05). The change in muscle activity as per the bar position, regardless of the load, significantly increased in all the muscles except BF during HBBS compared to that during LBBS (*p* < 0.05).

### 3.3. Muscle Activity of Trunk

Changes in trunk muscle activity according to the load and bar position are presented in Table 4 and Table 5, respectively. HBBS showed a significant increase in the muscle activity of all muscles in 70% compared to that in 60% in the eccentric phase (*p* < 0.05). In the concentric phase, the muscle activity of the RA was significantly increased in 60% compared to that in 50%, and other muscles significantly increased muscle activity in 70% compared to that in 60% (*p* < 0.05).

**Table 2 ijerph-19-13480-t002:** Comparison of the muscle activity of the lower extremity during the eccentric phase.

Muscle	50%	60%	70%	*p*-Value	Post Hoc Analysis		
					50% vs. 60%	50% vs. 70%	60% vs. 70%
VM							
HBBS	33.82 ± 12.30	36.57 ± 13.15	41.49 ± 14.28	<0.001	0.031	<0.001	<0.001
LBBS	31.13 ± 9.58	34.07 ± 12.50	36.03 ± 11.78	<0.001	0.021	<0.001	0.088
*p*-value	0.011	0.001	<0.001				
VL							
HBBS	30.95 ± 11.93	35.64 ± 14.23	38.93 ± 15.73	<0.001	<0.001	<0.001	0.001
LBBS	28.68 ± 11.53	31.86 ± 12.72	34.13 ± 14.78	<0.001	0.007	<0.001	0.064
*p*-value	0.006	<0.001	<0.001				
RF							
HBBS	20.54 ± 9.95	23.60 ± 12.57	26.64 ± 13.41	<0.001 ^†^	0.004 *	<0.001 *	0.007 *
LBBS	17.04 ± 9.31	19.74 ± 10.46	20.46 ± 9.67	<0.001 ^†^	0.002 *	0.002 *	0.245 *
*p*-value	0.002 *	<0.001 *	<0.001 *				
BF							
HBBS	17.77 ± 11.14	19.72 ± 12.63	20.98 ± 13.48	<0.001	<0.001	0.001	0.085
LBBS	16.76 ± 10.10	18.26 ± 11.18	19.79 ± 12.27	<0.001	0.028	<0.001	0.008
*p*-value	0.047	0.001	0.080				

Note. ^†^ Friedman test, * Wilcoxon signed-rank test. Abbreviation. VM, vastus medialis; VL, vastus lateralis; RF, rectus femoris; BF, biceps femoris; HBBS, high bar back squat; LBBS, low bar back squat.

**Table 3 ijerph-19-13480-t003:** Comparison of the muscle activity of the lower extremity during the concentric phase.

Muscle	50%	60%	70%	*p*-Value	Post Hoc Analysis		
					50% vs. 60%	50% vs. 70%	60% vs. 70%
VM							
HBBS	39.02 ± 11.33	42.46 ± 13.70	46.23 ± 10.76	<0.001	0.017	<0.001	0.040
LBBS	35.03 ± 9.87	37.51 ± 11.84	41.19 ± 11.69	<0.001	0.036	<0.001	<0.001
*p*-value	<0.001	<0.001	<0.001				
VL							
HBBS	36.74 ± 14.12	41.62 ± 17.65	45.44 ± 18.68	<0.001	0.003	<0.001	0.015
LBBS	32.70 ± 12.71	36.31 ± 15.11	40.21 ± 18.13	<0.001	0.001	<0.001	0.007
*p*-value	<0.001	<0.001	<0.001				
RF							
HBBS	24.27 ± 11.43	27.22 ± 13.87	30.66 ± 12.77	<0.001	0.059	0.001	0.121
LBBS	20.47 ± 11.72	21.94 ± 10.59	26.17 ± 11.76	<0.001 ^†^	0.430 *	0.001 *	0.002 *
*p*-value	<0.001 *	<0.001	0.001				
BF							
HBBS	21.07 ± 11.22	24.00 ± 13.34	28.06 ± 16.01	<0.001	0.001	<0.001	0.003
LBBS	20.58 ± 10.39	23.68 ± 12.77	26.15 ± 13.14	<0.001	0.001	<0.001	0.002
*p*-value	0.484	0.670	0.033				

Note. ^†^ used Friedman test, * used Wilcoxon signed-rank test. Abbreviation. VM, vastus medialis; VL, vastus lateralis; RF, rectus femoris; BF, biceps femoris; HBBS, high bar back squat; LBBS, low bar back squat.

**Table 4 ijerph-19-13480-t004:** Comparison of the muscle activity of the trunk during the eccentric phase.

Muscle	50%	60%	70%	*p*-Value	Post Hoc Analysis		
					50% vs. 60%	50% vs. 70%	60% vs. 70%
RA							
HBBS	2.08 ± 1.64	2.28 ± 1.91	2.63 ± 2.05	<0.001 ^†^	0.151 *	<0.001 *	0.021 *
LBBS	2.37 ± 2.11	2.50 ± 2.16	2.72 ± 2.05	0.019 ^†^	0.158 *	0.010 *	0.042 *
*p*-value	0.084 *	0.044 *	0.316 *				
TrA							
HBBS	8.52 ± 5.38	9.03 ± 5.61	11.75 ± 8.21	<0.001 ^†^	0.345 *	<0.001 *	0.002 *
LBBS	9.76 ± 6.72	13.83 ± 19.64	15.07 ± 18.47	0.002 ^†^	0.034 *	0.003 *	0.179 *
*p*-value	0.210 *	0.015 *	0.210 *				
EO							
HBBS	8.16 ± 4.95	8.66 ± 4.67	10.41 ± 5.98	<0.001 ^†^	0.210 *	<0.001 *	0.001 *
LBBS	9.31 ± 6.61	10.22 ± 6.76	11.19 ± 6.19	0.001 ^†^	0.012 *	0.003 *	0.092 *
*p*-value	0.295 *	0.068 *	0.040 *				
ES							
HBBS	21.00 ± 11.27	22.60 ± 11.08	28.47 ± 14.53	<0.001	0.806	<0.001	0.002
LBBS	25.23 ± 13.46	29.63 ± 14.63	31.15 ± 13.25	0.002	0.001	0.009	0.949
*p*-value	<0.001	0.001	0.085				

Note. ^†^ used Friedman test, * used Wilcoxon signed-rank test. Abbreviation. RA, rectus abdominis; TrA, transverse abdominis; EO, external oblique; ES, erector spinae; HBBS, high bar back squat; LBBS, low bar back squat.

**Table 5 ijerph-19-13480-t005:** Comparison of the muscle activity of the trunk during the concentric phase.

Muscle	50%	60%	70%	*p*-Value	Post Hoc Analysis		
					50% vs. 60%	50% vs. 70%	60% vs. 70%
RA							
HBBS	2.43 ± 1.74	2.77 ± 1.96	3.22 ± 2.14	<0.001 ^†^	0.010 *	<0.001 *	0.101 *
LBBS	2.87 ± 2.05	3.32 ± 2.00	3.39 ± 2.22	0.005 ^†^	0.031 *	0.025 *	0.699 *
*p*-value	0.013 *	0.004 *	0.080 *				
TrA							
HBBS	11.62 ± 9.47	11.84 ± 7.83	15.86 ± 11.54	<0.001 ^†^	0.311 *	<0.001 *	0.001 *
LBBS	13.96 ± 11.87	15.39 ± 11.72	17.41 ± 13.53	<0.001 ^†^	0.008 *	<0.001 *	0.255 *
*p*-value	0.179 *	0.001 *	0.068 *				
EO							
HBBS	10.11 ± 7.23	10.62 ± 6.56	13.44 ± 9.28	<0.001 ^†^	0.284 *	<0.001 *	0.001 *
LBBS	10.24 ± 6.57	12.26 ± 8.64	14.76 ± 10.35	<0.001 ^†^	0.001 *	<0.001 *	0.002 *
*p*-value	0.187 *	0.088 *	0.064 *				
ES							
HBBS	20.84 ± 11.37	21.16 ± 9.55	26.58 ± 14.51	<0.001 ^†^	0.246 *	<0.001 *	<0.001 *
LBBS	22.85 ± 11.61	26.45 ± 14.56	30.75 ± 12.63	<0.001 ^†^	<0.001 *	<0.001 *	0.003 *
*p*-value	0.036 *	0.004 *	0.043 *				

Note. ^†^ used Friedman test, * used Wilcoxon signed-rank test. Abbreviation. RA, rectus abdominis; TrA, transverse abdominis; EO, external oblique; ES, erector spinae; HBBS, high bar back squat; LBBS, low bar back squat.

On performing the LBBS, the activity of all muscles increased significantly as the load increased. In the eccentric phase, the RA showed a significant change ranging from 60% to 70%, and other muscles showed a significant increase between 50% and 60% (*p* < 0.05). In the concentric phase, RA and TrA showed significant changes between 50% and 60%, whereas EO and ES muscles showed a significant increase between 60% and 70% (*p* < 0.05). With respect to the change in muscle activity according to bar position, only ES showed a significant increase in HBBS compared to that in LBBS, regardless of the load (*p* < 0.05).

## 4. Discussion

In this study, the following objectives were set to investigate the effect of load and bar position on muscle activity in the trunk and lower extremities during a squat exercise. (1) The muscle activity of the trunk and lower extremities changes according to the difference in bar position (HBBS and LBBS) during a squat. (2) As the load increases during a squat, the muscle activity of the trunk and lower extremities increases. The results of this study confirmed that the changes in load and bar position during a squat induced significant changes in the muscle activity of the trunk and lower extremities.

The sEMG can detect and quantify the muscle activity required to perform movements. In the absence of clear information on the muscle activity of the trunk and lower extremities required for back squat performance, the results of this study would be helpful for medical doctors, physical therapists, and athletic trainers who need to prescribe the appropriate method and amount of squat exercise.

We measured the muscle activities of the lower extremities in the VM, VL, RF, and BF muscles. HBBS showed significantly higher muscle activity than LBBS in the VM, VL, and RF in all the measurements. In contrast, in BF, HBBS showed significantly higher muscle activity than LBBS in 50% and 60% loads in the eccentric phase and 70% load in the concentric phase (*p* < 0.05; Table 2 and Table 3). In HBBS, the bar is placed on top of the trapezius just below the spinous process of the 7th cervical vertebrae (C7) [15]. Owing to the position of the bar, HBBS allows a relatively deeper squat by inducing more knee flexion than LBBS [29,30]. The increase in knee flexion caused by a larger range of squat motion further increases the muscle activity of the lower extremities [31]. Therefore, HBBS caused a higher activity of lower extremity muscles by providing a larger range of motion than LBBS in this study.

In a previous study, the BF muscle activity was relatively lower than that of the RF and ES during squats [32]. Similarly, in our results, the BF showed lower activity than other lower extremity muscles in both the concentric and eccentric phases (Table 2 and Table 3). In addition, according to a study comparing the absolute muscle activity between HBBS and LBBS, the muscle activity of the gluteus and hamstring musculature in the sticking region was lower than that of the quadriceps musculature [33]. Based on these results, it is presumed that quadriceps, such as the VM, VL, and RF, have a greater effect on squat performance than hamstrings.

In trunk muscles, LBBS showed a significantly higher muscle activity than HBBS in the RA, TrA, and EO, except in a few cases (RA, 50% load in concentric phase and 60% load in all phases; TrA, 60% load in all phases; EO, 70% load in eccentric phase and 70% load in concentric phase), and ES showed a significantly higher muscle activity in all measurements in this study.

To perform LBBS, the participant places a bar on the lower trapezius just above the posterior deltoid along the scapular spine [15]. Compared with HBBS, LBBS induces a relatively larger anterior tilt of the pelvis by the position of the bar and also causes the participant’s trunk to tilt more anteriorly, increasing the load on the hip joint rather than the knee joint [23,34]. This posture induces a greater muscle activity of the ES and increases the co-contraction with the trunk flexors, providing trunk neutrality and stability [35]. However, the EO did not show a significantly higher muscle activity than the HBBS, except for a 70% load in the eccentric phase. These results are thought to be due to differences in the anatomical structure of the EO and other abdominal muscles and their functions. The EO is a muscle located on the side of the abdomen and is mainly involved in the lateral flexion and rotation of the trunk. Therefore, it is thought that the EO, a muscle that mainly acts in the movement to control the frontal plane, is less affected by the change in the sagittal plane due to the bar position in the squat movement, which mainly requires the adjustment force in the sagittal plane [36]. Our results demonstrated that LBBS was more effective than HBBS in inducing greater activation of ES, RA, and TrA. In addition, we propose that LBBS training at a load of 60% is the most effective application.

Contrary to our results, according to a study by Van den Tillaar et al. (2020) that compared the difference in muscle activity between the lower extremity and erector muscles in HBBS and LBBS, ES showed higher activity in HBBS than in LBBS [33]. Unlike our study, this report showed no difference in the forward tilt of the trunk between the two conditions. As the change in the moment arm caused by the forward tilt of the trunk disappeared, only the change in the moment arm caused by the height of the bar must have affected the activity of the ES. Therefore, the results of this study appropriately identified the effect of the back squat. Several rehabilitation centers and gyms perform movements using this method.

In our study, the muscle activity of all muscles of the trunk and lower extremities significantly increased between a load of 50% and 70%, regardless of the bar position and phase. In the HBBS, there was no significant difference in muscle activity between 60% and 70% of load of BF and RF among the muscles of the lower extremities, between 50% and 60% of load in the eccentric phase of all muscles of the trunk, between 60% and 70% in the concentric phase of RA, and between 50% and 60% of TrA, EO, and ES muscles. In LBBS, there was no significant difference between 60% and 70% of load in the eccentric phase of VM, VL, and RF and between 50% and 60% in the concentric phase of RF among the lower extremity muscles. In the trunk muscle, no significant change was observed in the activity of RA (between 50% and 60%) and TrA, EO, and ES (between 60% and 70%) in the eccentric phase and between RA and TrA (between 60% and 70%) in the concentric phase.

McCaw’s study tested changes in muscle activity according to the load exerted during squats and reported that the muscle activity increased with increasing load [23]. In addition, Paoli et al. (2009) studied the changes in muscle activity according to squat motions for loads of 0%, 30%, and 70% in professional lifters, and found that muscle activity increased with an increase in load [37], consistent with our study findings and McCaw’s results. In particular, the comparison between 50% and 70% in this study showed that the activities of all muscles were significantly increased. Based on these results, we can speculate that the amount of change in load should be at least 20% or more for higher-intensity training of the lower extremities and trunk muscles during squat exercises.

This study had some limitations. First, in order to evaluate trunk stability, it was not measured directly using a specific instrument but indirectly through the measurement of muscle activity. Second, in this study, since our sEMG system could measure only up to 8 channels, only eight muscles of the trunk and lower extremity related to the squat were evaluated. The activity of other trunk muscles, lower extremity, and upper extremity muscles related to the squat is unclear. Third, the subjects of this study were all males aged 20–40 years. The results of this study cannot be generalized to other age groups and women. Finally, the small sample size may also affect the results of this study.

## 5. Conclusions

In this study, HBBS showed high activity in the lower extremity muscles, whereas LBBS showed high activity in the trunk muscles. In addition, as the load increased during squatting, the muscle activity in the trunk and lower extremities increased. We recommend that rehabilitation experts, such as physical therapists and athletic trainers, apply the bar position and load as variables affecting the intensity and method of training to strengthen the target muscles.

## Figures and Tables

**Figure 1 ijerph-19-13480-f001:**
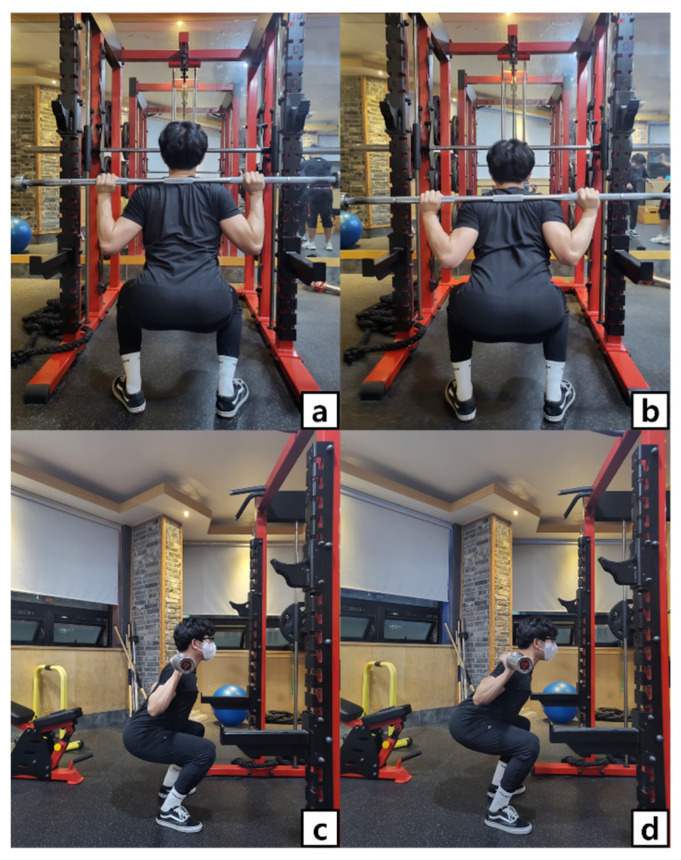
Bar positioning during barbell squat. (**a**) posterior view of HBBS, (**b**) posterior view of LBBS, (**c**) lateral view of HBBS, (**d**) lateral view of LBBS.

**Table 1 ijerph-19-13480-t001:** Characteristics of participants.

Variables	
Age (year)	28.71 ± 3.10
Height (m)	1.74 ± 0.06
Weight (kg)	75.00 ± 12.90
BMI (kg/m^2^)	24.66 ± 3.53
Training experience (year)	1.82 ± 0.82
1 RM (kg)	101.43 ± 14.84

Abbreviation. BMI, body mass index; 1 RM, one repetition maximum.

## Data Availability

Not applicable.

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
