# Peer review of "Influence of Loads and Loading Position on the Muscle Activity of the Trunk and Lower Extremity during Squat Exercise"

_ijerph, 2022, doi:10.3390/ijerph192013480_

Round 1
Reviewer 1 Report
I appreciate your effort toward "Influence of loads and loading position on the muscle activity of the trunk and lower extremity during squat exercise"
Comments and Suggestions:
1. Page 1, Line Line 22: What test showed a significant difference?
2. Page 1, Line Line 29: "Need of the study" is missing. Also, the study is limited in the generalization of findings/results.
3. Page 2, Line 69-75: I failed to understand the importance of the study. The research gap is described in previous studies. However, the need for the study is not explained.
4. Page 2, Line 80-81: The study location is not described properly.
5. Page 2, Line 81: Justification for selecting a specific age group (20-40) is missing.
6. Page 3, Line 101-102: Required literature support or reference
7. Page 3, Line 97-111: "Study procedure/protocol" Justification required along with literature. Perhaps several questions were raised like why 3 days? why 10 mint? why not 12 mint....
8. Page 3, Line 108-111: How these muscles were determined or selected?
9. Page 3, Line 125: Suggested to include subject's pictures which demonstrate the experiments
10. Page 4, Line 172: SI units should be followed. There is no consistency. For example, Height units are in centimeters and BMI in (kg/m^3).
11. Page 3, Line 98: "Training experience" is missing
12. Page 4, Line 164: Why "Bonferronicorrrection" was used for posthoc testing?
13. Page 4, Line 166: The significance levels have to be mentioned as "(p < 0.05)". This correction has to be done throughout the manuscript.
14. Page 8, 298-299: This is very obviously true. However, the author should state other gender-specific limitations in the study. For example smaller sample size of males, age group (20-40), etc...
Author Response
I would like to thank you for providing the opportunity to revise and resubmit the attached manuscript entitled “Influence of loads and loading position on the muscle activity of the trunk and lower extremity during squat exercise” for publication in IJERPH.
We deeply appreciate the editorial comments and reviewers’ helpful comments on our manuscript which we ignored. We agreed with the points addressed by the Reviewers. We provide our responses to the Reviewers’ comments. Please review the attached files.

Reviewer 2 Report
Dear Authors,
Thank you for the opportunity to review your manuscript. The study of muscular activation of different squat positions is interesting. Nevertheless, some parts of your manuscript are unclear, which I would like to address in the following.
Line 19: „…was measured by means of surface EMG.“
Line 24-26: unclear – in relation to what? Please rephrase.
Line 26-28: „requires more force…“ – you did not measure forces, you can only make statements about muscle activity.
Line 30: „Lower extremity“
Line 47: please specifiy what you understand under „core muscles“.
Line 81: „Incheon“, please add country
Line 95: what happened to the missing 4 participants?
Line 97 ff: „Informed consent statement“ is missing.
Line 108: „Squats were then performed on the vastus medialis (VM), …“ I do not understand this sentence. Do you mean that sEMG measurements were performed on the muscles?
Line 108: Figure 1 is missing in the manuscript.
Line 116: do you mean „Scapulae“ or „Shoulderblades“?
Line 129: I do not understand: did you attach two electrodes – first a disposable electrode and then another? Please check sentence.
Line 122: Are 2 minutes rest enough to regenerate muscle power (please give a reference)?
Line 128: „AcqKnowledge“
Line 137: „SENIAM“
Line 150 & 151: bending (flexion) and extension are interchanged.
Line 153: flexion instead of extension
Line 172 – Table 1: Check the footnote! (a) and (b) without reference in the table.
Line 173 ff: increased in comparision to what? (resting state / 1 repetition maximum?)
The following paragraphs are unclear. Please be more precise: did a muscle activity significantly increase by 70% (meaning it became 70% larger than before) or did it increase under the 70%-of-1RM-loading condition?
Line 229: cervical vertebrae, not spine
Line 239: activity of the gluteus and hamstring musculature
Line 264: Van den Tillaar
Line 266: „Unlike our study, this report showed no difference in the forward tilt of the trunk …“ (How) did you measure the forward tilt oft he trunk in your study?
Line 296: I would proprose: „since our sEMG system could measure only up to …“
Line 322: Please add the country.
Author Response

(The authors gave the same response as above.)

Round 2
Reviewer 1 Report
I appreciate the author's effort toward the revised manuscript. Recommended for publication